# A network pharmacology approach to determine the underlying mechanisms of action of Yishen Tongluo formula for the treatment of oligoasthenozoospermia

Yangdi Chen[1], Fanggang Bi[2], Zixue Sun[3]*

1 Henan University of Chinese Medicine, Zhengzhou, Henan, P. R. China, 2 Department of Orthopedic Surgery, The First Affiliated Hospital of Zhengzhou University, Zhengzhou, P. R. China, 3 Department of Reproductive Medicine, Henan Province Hospital of Traditional Chinese Medicine (The Second Affiliated Hospital of Henan University of Chinese Medicine), Zhengzhou, Henan, P. R. China

* sunhhzx@163.com

**Data Availability Statement:** All relevant data are within the paper.

## Abstract

Oligoasthenozoospermia is a complex disease caused by a variety of factors, and its incidence is increasing yearly worldwide. Yishen Tongluo formula (YSTLF), created by Professor Sun Zixue, has been used to treat oligoasthenozoospermia in clinical practice for several decades with a good therapeutic effect. However, the chemical and pharmacological profiles of YSTLF remain unclear and need to be elucidated. In this study, a network pharmacology approach was applied to explore the potential mechanisms of YSTLF in oligoasthenozoospermia treatment. All of the compounds in YSTLF were retrieved from the corresponding databases, and the bioactive ingredients were screened according to their oral bioavailability (OB) and drug-likeness (DL). The potential proteins of YSTLF were obtained from the traditional Chinese medicine systems pharmacology (TCMSP) database and the Bioinformatics Analysis Tool for Molecular Mechanism of Traditional Chinese Medicine (BATMAN-TCM) database, while the potential genes of oligoasthenozoospermia were obtained from the GeneCards database and the DisGeNET database. The STRING database was used to construct an interaction network according to the common targets identified by the online tool Venny for YSTLF and oligoasthenozoospermia. The topological characteristics of nodes were visualized and analyzed through Cytoscape. Biological functions and significant pathways were determined and analyzed using the Gene Ontology (GO) knowledgebase, the Kyoto Encyclopedia of Genes and Genomes (KEGG) and Metascape. Finally, the disease-formula-compound-target-pathway network was constructed by Cytoscape. A total of 106 bioactive ingredients and 134 potential targets from YSTLF were associated with oligoasthenozoospermia or considered to be therapeutically relevant. Pathway analysis indicated that the PI3K/Akt, MAPK and apoptosis signaling pathways were significant pathways involved in oligoasthenozoospermia. In conclusion, the current study expounded the pharmacological actions and molecular mechanisms of YSTLF in treating oligoasthenozoospermia from a holistic viewpoint. The potential molecular mechanisms were closely related to antioxidative stress, antiapoptosis and anti-inflammation, with TNF,

**Funding:** This study was funded by the National Natural Science Foundation of China (NSFC) (81974573), Central Plains Thousand Talents Program of Henan Province (ZYMY201809) –Central Plains Famous Doctors Project and Major Special Project of Henan Province for Chinese Medical Researches (2018ZYZD10).

**Competing interests:** The authors have declared that no competing interests exist.

CCND1, ESR1, NFKBIA, NR3C1, MAPK8, and IL6 being possible targets. This network pharmacology prediction may offer a helpful tool to illustrate the molecular mechanisms of the Chinese herbal compound YSTLF in oligoasthenozoospermia treatment.

## Background

In recent years, the incidence of male infertility has increased each year due to the influences of environmental pollution, psychological hazards, drug abuse, unhealthy living habits and other factors. According to the World Health Organization (WHO), infertility currently affects nearly 15% of couples of childbearing age worldwide, with male factors accounting for more than 50% of these cases [1, 2]. Infertility, which seriously threatens human reproductive health, has become the third most difficult disease in the world, following only cardiovascular disease and cancer [3]. Among the many causes of male infertility, reduced sperm count and decreased sperm motility are two of the most common. The WHO has stated that asthenospermia is defined as having a rapid forward motile sperm (PR) of less than 32% or a total of forward motile sperm and nonforward motile sperm (PR+NP) of less than 40% within one hour after ejaculation with a semen density greater than $15\times10^6$/ml. Moreover, semen density less than $15\times10^6$/ml is considered oligozoospermia [4]. At present, there is no effective drug for the treatment of oligoasthenozoospermia (oligozoospermia combined with asthenospermia) in modern medicine. In recent years, a boom in assisted reproductive technology (ART) has solved certain fertility problems; however, there are still some defects, such as adverse reactions, genetic risk, high cost, and a low success rate [5]. Traditional Chinese medicine (TCM), under the guidance of holistic concepts and treatment based on syndrome differentiation, offers satisfactory therapeutic methods for the treatment of oligoasthenozoospermia with low adverse effects.

Yishen Tongluo formula (YSTLF), an empirical formula, was created by Professor Sun Zixue, a director of reproductive medicine department of Henan Provincial Hospital of Traditional Chinese Medicine. YSTLF is composed of seven herbs, including Semen Cuscutae (Tu Si Zi; TSZ), Herba Epimedii (Yin Yang Hou; YYH), Radix Rehmanniae Preparata (Shu Di Huang; SDH), Radix Astragali (Huang Qi; HQ), Salvia Miltiorrhiza (Dan Shen; DS), Leech (Shui Zhi; SZ), and Radix Cyathulae (Chuan Niu Xi; CNX). Previous clinical trials have demonstrated that YSTLF can significantly improve semen density, a/a+b or PR/PR+NP sperm, and sperm motility (total effective rate: 89.74% vs 64.10% (p<0.05)) [6]. Despite the above findings, the molecular mechanisms underlying the therapeutic effects of YSTLF on oligoasthenozoospermia remain unclear. Thus, further research with the appropriate approaches is warranted to comprehensively reveal the involved potential mechanisms.

With a surge of progress in bioinformatics, network pharmacology has opened up a new field of pharmacological research. Based on the theories of multidirectional pharmacology and systems biology, network pharmacology can construct complex network models to study the biological or pharmacological properties of a target and explore its physiological or pharmacological mechanism supported by high-throughput data analysis, virtual computing technology and network public databases, etc [7, 8]. The systematic and holistic characteristics of network pharmacology correspond with the core ideas of the holistic philosophy of traditional Chinese medicine (TCM). Network pharmacology can be applied to illustrate the interactive relationship between multiple components, targets and pathways of bioactive compounds in TCM herbal medicines [9], which could help to evaluate the compatibility and rationality of Chinese medicinal formulae. Therefore, an increasing number of studies have depended on network

pharmacology to probe the potential targets and molecular mechanisms of TCM herbal medicines on holistically complex diseases. Zhang et al. [10] relied on network pharmacology analysis and reported that Bushen Tiansui formula might treat Alzheimer's disease mainly through the TNF and PI3K/Akt signaling pathways. Lu et al. [11] demonstrated that Xijiao Dihuang Decoction (25 g/kg) could improve survival after sepsis by regulating the NF-κB and HIF-1α signaling pathways based on network pharmacology analysis.

Therefore, in this study, public database resources and computational tools were used to investigate the pharmacological network of YSTLF in oligoasthenozoospermia by a network pharmacology approach. The research purpose was to elucidate the potential mechanism of YSTLF in the treatment of oligoasthenozoospermia through a prediction of the bioactive ingredients and a bioinformatics analysis of common targets. The detailed network pharmacology strategy of the current study is presented in **Fig 1**.

## Materials and methods

### YSTLF bioactive ingredients

All of the compounds of each herb in YSTLF were retrieved from the traditional Chinese medicine systems pharmacology (TCMSP) database (http://lsp.nwu.edu.cn/tcmsp.php) and the Bioinformatics Analysis Tool for Molecular Mechanism of Traditional Chinese Medicine (BATMAN-TCM) database (http://bionet.ncpsb.org/batman-tcm/) [12, 13]. During the screening for active ingredients, the absorption, distribution, metabolism, and excretion (ADME) of drug compounds were the essential limitations. Oral bioavailability (OB) and drug-likeness (DL) are two of the most crucial pharmacokinetic parameters in ADME processes [14]. OB refers to the speed and extent of absorption of an orally administered drug that enters the body's circulation through the liver after absorption from the gastrointestinal tract. DL is defined as the structural similarity of an herbal ingredient to a known drug. Compounds with an OB≥30% and a DL≥0.18 were screened as bioactive ingredients in the current study [15, 16].

### YSTLF compound targets

The putative targets of the bioactive ingredients in YSTLF were obtained from the TCMSP database and the BATMAN-TCM database. The targets downloaded from the TCMSP database were converted from the protein name to the gene name via R language. All gene targets of YSTLF were retrieved after summarizing and deleting duplicates.

### Oligoasthenozoospermia gene targets

The keywords 'asthenospermia', 'oligospermia', 'low sperm motility', 'spermatogenic dysfunction' and 'deficiency of sperm motility' were used in the GeneCards database (https://www.genecards.org/) and the DisGeNET database (http://www.disgenet.org/) to collect oligoasthenozoospermia-related targets. The GeneCards database provides comprehensive, user-friendly information on all annotated and predicted human genes [17]. The DisGeNET database is a discovery platform containing one of the largest publicly available collections of genes and variants associated with human diseases [18]. The intersection between the putative target genes of YSTLF and oligoasthenozoospermia-related targets was determined with the online tool Venny 2.1 (http://bioinfogp.cnb.csic.es/tools/venny/) and these were considered the potential targets for the bioactive ingredients in YSTLF to treat oligoasthenozoospermia, which was visualized with a Venn diagram.

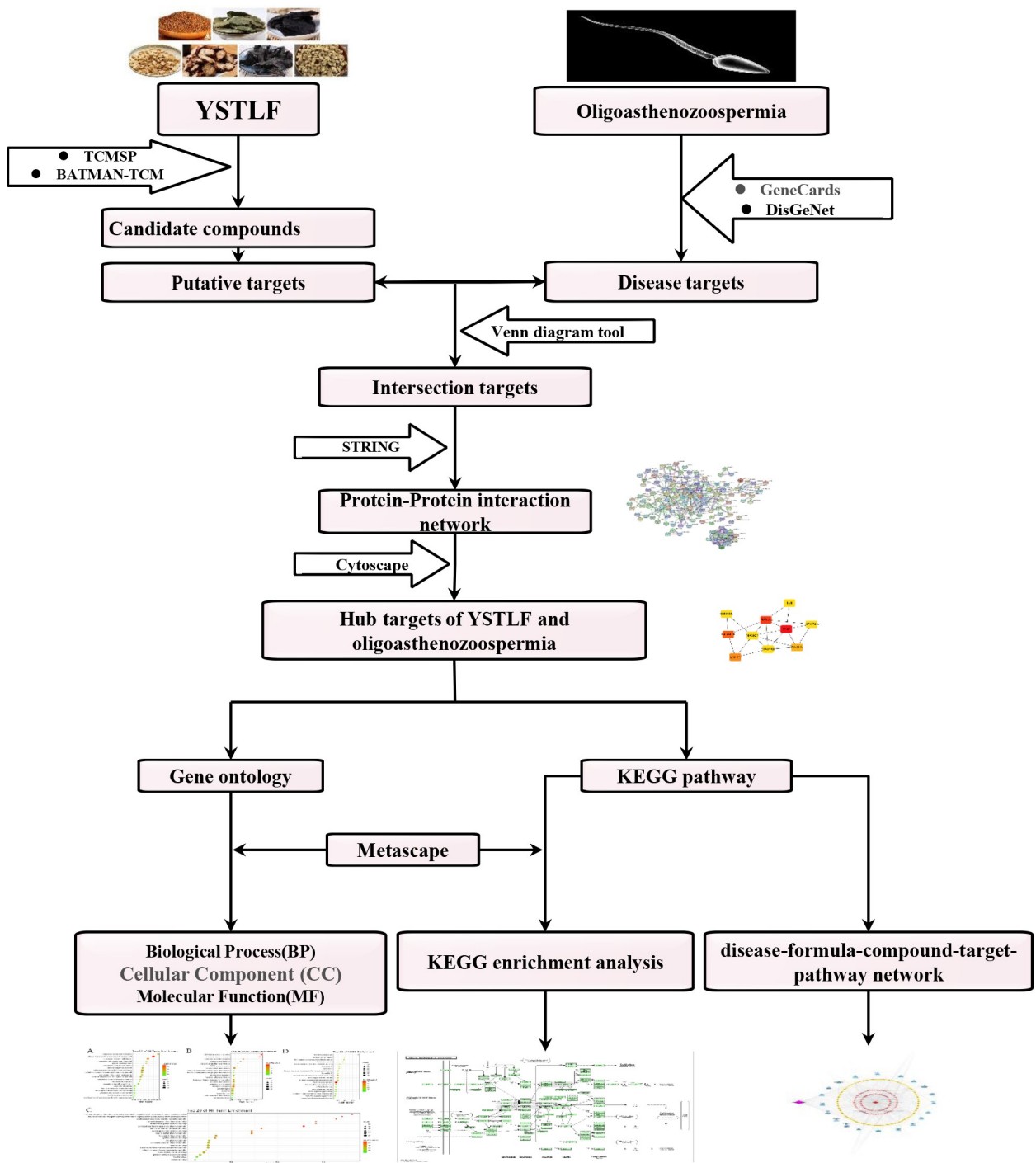

**Fig 1. Workflow for elucidation of YSTLF in the treatment of oligoasthenozoospermia.**

## Protein-protein interaction (PPI) network analysis

The common targets of YSTLF and oligoasthenozoospermia were put into the STRING database (https://string-db.org/, version 11.0) for PPI network analysis. The interacting proteins with a confidence score of ≥0.900 were chosen for PPI network visualization construction

[19]. Then, the constructed PPI network was displayed in Cytoscape software (version 3.7.2) [20], which was based on the information from the STRING database. The CytoHubba plugin (version 1.0) of Cytoscape was applied to compute the top ten genes with a high degree [21]. It is generally believed that the node with the highest degree, highest betweenness centrality and highest closeness centrality is the most topologically important in the network. In other words, the top ten genes were potential hub targets.

### GO and KEGG pathway analyses

The Gene Ontology (GO) knowledgebase can provide information on the functions of genes, including biological processes (BPs), cellular components (CCs) and molecular functions (MFs) [22]. The Kyoto Encyclopedia of Genes and Genomes (KEGG) is a database resource for understanding high-level gene functions and genomic information [23]. GO and KEGG pathway enrichment analyses were conducted in Metascape (http://metascape.org/), a gene annotation and analysis resource [24]. According to a high count and P<0.05, the top 20 BP, CC, and MF GO terms and the top 20 KEGG pathways were selected for functional annotation clustering to construct enrichment analysis bubble diagrams by the clusterProfiler package in R [25]. Enrichment analyses could contribute to further probes of the biological functions and potential mechanisms of the YSTLF targets in oligoasthenozoospermia.

### Construction of the networks and analysis

To further clarify the molecular mechanism of YSTLF in oligoasthenozoospermia treatment, Cytoscape 3.7.2 was applied to establish a disease-formula-compound-target-pathway network [26]. In this graphical network, the disease, formula, compounds, targets and pathways were expressed as nodes, whereas the disease-formula-compound-target-pathway interactions were expressed as edges.

## Results

### YSTLF bioactive components

Based on the OB and DL values, 130 active components were finally obtained by searching the TCMSP database and the BATMAN-TCM database. Specifically, the numbers of candidate ingredients in TSZ, YYH, SDH, HQ, DS, CNX and TSZ were 11, 23, 2, 20, 65, 4, and 15, respectively. Among all ingredients, 7 involved common compounds from two or more drugs. For example, quercetin is a common compound in the four drugs TSZ, YYH, HQ, and CNX, and it has antioxidant, anti-inflammatory, antiviral, and antitumor activities and regulates glucose and lipid metabolism, and immune functions [27].

### Potential targets of YSTLF for oligoasthenozoospermia

Among the 130 candidate components, 955 targets were retrieved from the TCMSP database and the BATMAN-TCM database. Finally, a total of 250 ingredient targets in YSTLF were obtained after eliminating the overlapping targets. The numbers of potential targets connected by TSZ, YYH, SDH, HQ, DS, CNX and TSZ were 108, 106, 21, 100, 61, 88, and 142, respectively.

According to previous literature, oligoasthenozoospermia is a genetic disease. Based on the GeneCards database and the DisGeNET database, a total of 3677 genes associated with oligoasthenozoospermia were obtained after deleting duplicates. Specifically, the numbers associated with 'asthenospermia', 'oligospermia', 'low sperm motility', 'spermatogenic dysfunction' and 'deficiency of sperm motility' were 145, 672, 3229, 771, and 2936, respectively.

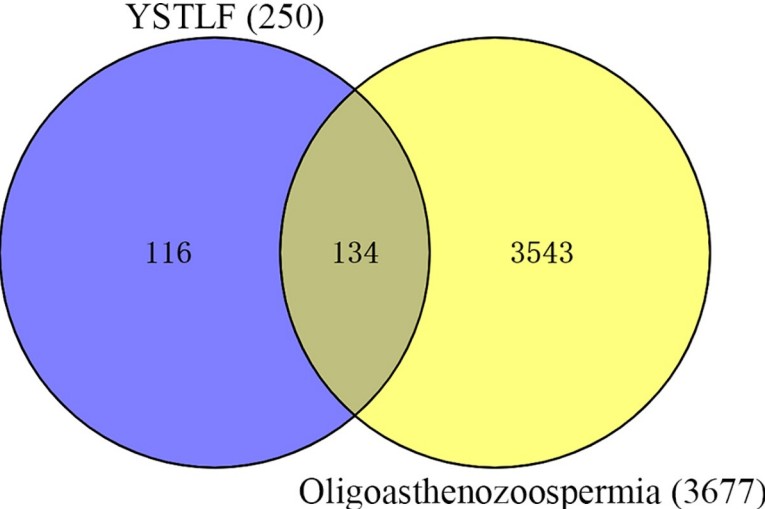

**Fig 2. Venn diagram of the drugs and disease targets.**

Then, the intersection between 250 YSTLF targets and 3677 oligoasthenozoospermia-related targets was determined through the online tool Venny 2.1. Consequently, a total of 134 potential targets associated with both oligoasthenozoospermia and YSTLF were identified with Venn diagrams (**Fig 2**).

## YSTLF compound-target network

As Chinese medicinal formulae exhibit multiple pharmacological effects by interacting with multiple targets, it is of great significance for us to explore the underlying mechanisms of Chinese medicinal formulae on complex diseases through network analysis. Cytoscape 3.7.2 was used to construct the compound-target network of YSTLF on oligoasthenozoospermia (**Fig 3**). Through network analysis in Cytoscape 3.7.2, there were 240 nodes (106 for candidate bioactive ingredients and 134 for potential targets) as well as 607 edges in the network. Based on the topological analysis of the network, the number of edges or targets related to the nodes is taken as the degree value. Among all the bioactive ingredients, the top 10 with the highest degree of nodes were quercetin, luteolin, kaempferol, crocetin, D-mannitol, ursolic acid, tanshinone iia, isorhamnetin, anhydroicaritin, and beta-sitosterol (**Table 1**), which were involved in the regulation of multiple oligoasthenozoospermia targets. The top 10 compounds mentioned above could be regarded as potentially the most bioactive core components. These findings may provide new mechanisms for the treatment of oligoasthenozoospermia by YSTLF.

## PPI network

Growing studies have confirmed that diseases are not caused by only a single gene but also via interactions among multiple targets. To further elaborate the molecular mechanisms of the pharmacological actions of YSTLF on oligoasthenozoospermia, 134 potential targets were input into the STRING database to construct a PPI network. In the network, the nodes and edges represent proteins and protein-protein associations, respectively. In the current study, 134 nodes and 580 edges were established in the network after hiding disconnected nodes (**Fig 4**). The average node degree and the average local clustering coefficient were 8.66 and 0.541, respectively. The top 10 hub genes (i.e., TNF, RELA, CCND1, ESR1, RXRA, NFKBIA, GSK3B, NR3C1, MAPK8, IL6), which were considered potential hub targets of YSTLF for

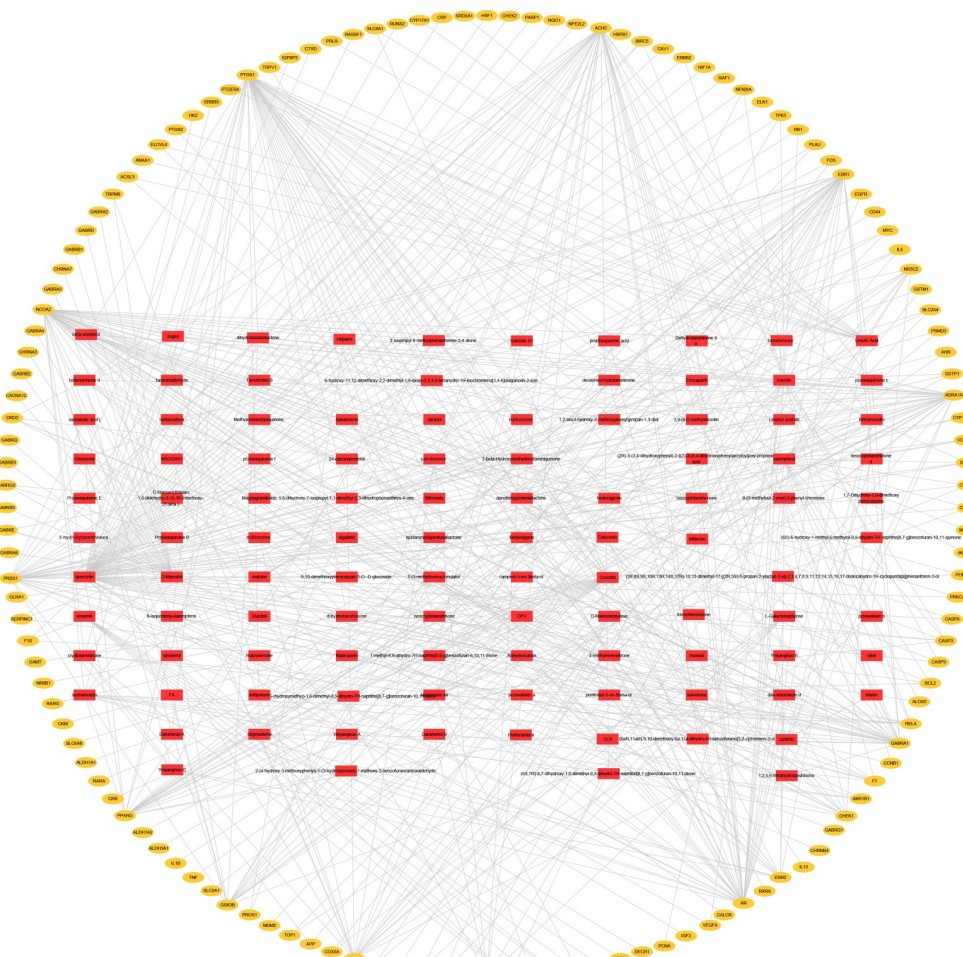

**Fig 3. Compound-target network of YSTLF.** Red rectangles represent the compounds in YSTLF, and orange circles represent the targets of oligoasthenozoospermia.

treating oligoasthenozoospermia, were identified through the CytoHubba plugin with maximal clique centrality (MCC) according to their degree (**Fig 5**). The network characteristics of the potential hub targets are presented in **Table 2**.

**Table 1. Top 10 bioactive ingredients with a high degree.**

| Mol ID | Molecule Name | Degree |
|---|---|---|
| MOL000098 | quercetin | 59 |
| MOL000006 | luteolin | 26 |
| MOL000422 | kaempferol | 25 |
| MOL001406 | crocetin | 22 |
| MOL000003 | D-mannitol | 21 |
| MOL000511 | ursolic acid | 17 |
| MOL007154 | tanshinone iia | 14 |
| MOL000354 | isorhamnetin | 14 |
| MOL004373 | anhydroicaritin | 11 |
| MOL000358 | beta-sitosterol | 11 |

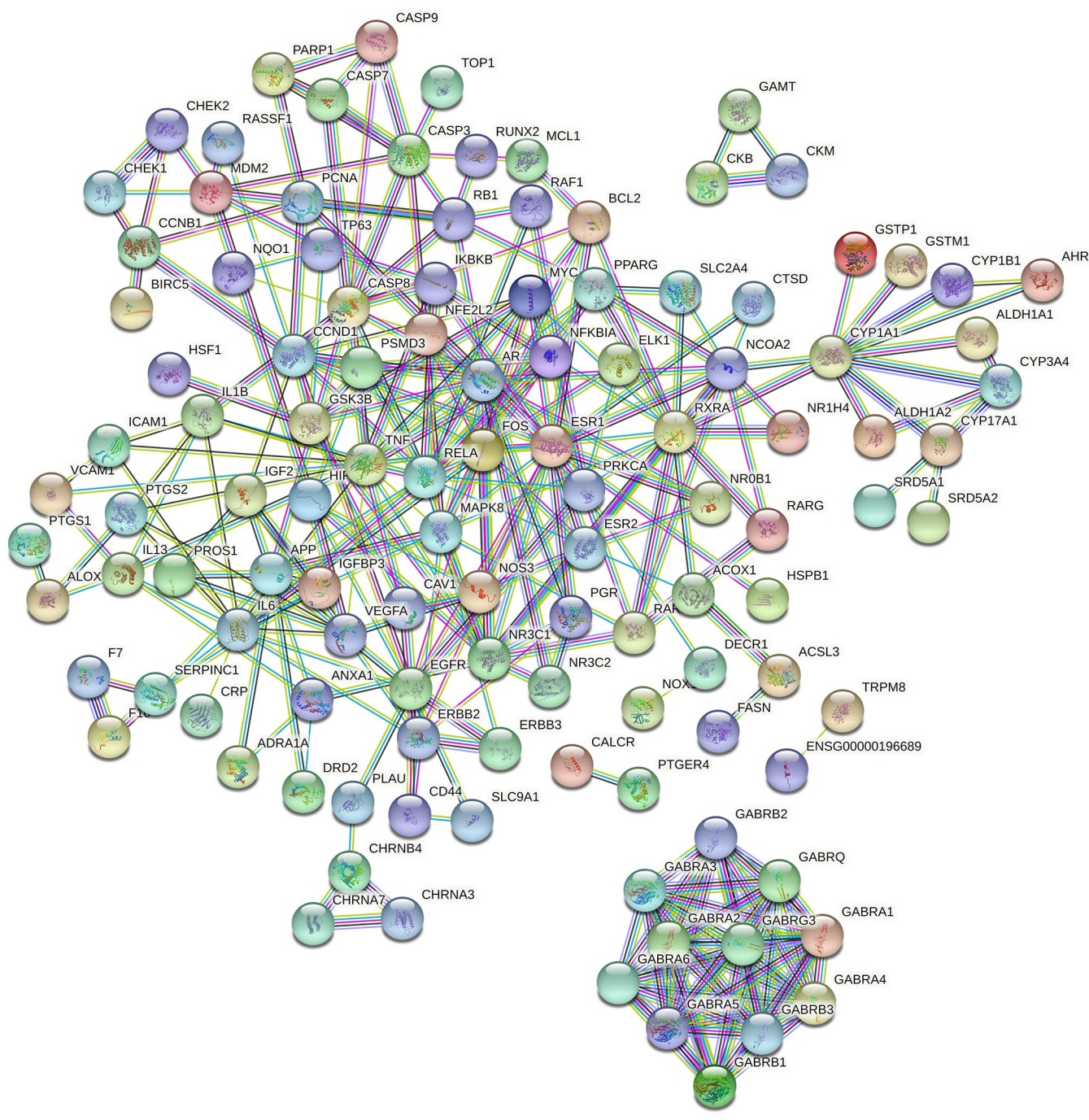

**Fig 4. Common target protein interaction network.**

Closeness and betweenness are two more important parameters in the network. Closeness refers to the reciprocal of the average length of the shortest circuit between a node and all other nodes in the network. The higher the value is, the greater the centrality of the node, indicating that the transmission speed of the signal from one node to the other nodes is faster. Betweenness is defined as the shortest path of all node pairs in the network, and the shorter the paths through a node are, the more important that node is.

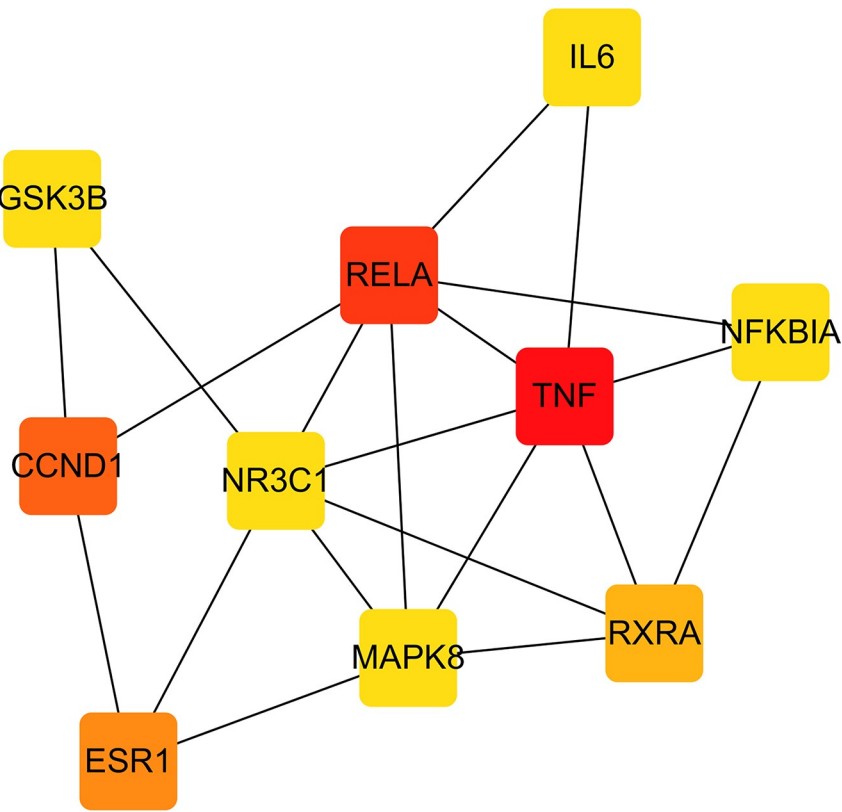

**Fig 5. Analysis of the top 10 hub gene networks of YSTLF for oligoasthenozoospermia treatment by the MCC algorithm, in which red and yellow represent the importance in the network.**

## GO terms enrichment analysis

To illustrate the multiple biological functions of potential targets in oligoasthenozoospermia with treatment by YSTLF, the above common targets were entered into Metascape for GO enrichment analysis. The results of the GO analysis showed that the biological process (BP) was significantly enriched in the cellular response to organic cyclic compounds (GO: 0071407), cellular response to hormone stimuli (GO: 0032870), response to toxic substances (GO: 0009636), cellular response to nitrogenous compounds (GO: 1901699), cellular response

**Table 2. The PPI network characteristics of the top 10 hub targets.**

| Target | Betweenness | Closeness | Degree |
|--------|-------------|-----------|--------|
| TNF | 1556.31697 | 58.56667 | 26 |
| RELA | 1260.83584 | 57.63333 | 24 |
| CCND1 | 1124.66339 | 54.35 | 19 |
| ESR1 | 1052.45654 | 52.48333 | 16 |
| RXRA | 2025.27701 | 53.31667 | 15 |
| NFKBIA | 220.14611 | 49.31667 | 14 |
| GSK3B | 392.20756 | 47.13333 | 14 |
| NR3C1 | 658.08329 | 52.4 | 14 |
| MAPK8 | 992.50018 | 54.06667 | 14 |
| IL6 | 851.07865 | 50.06667 | 14 |

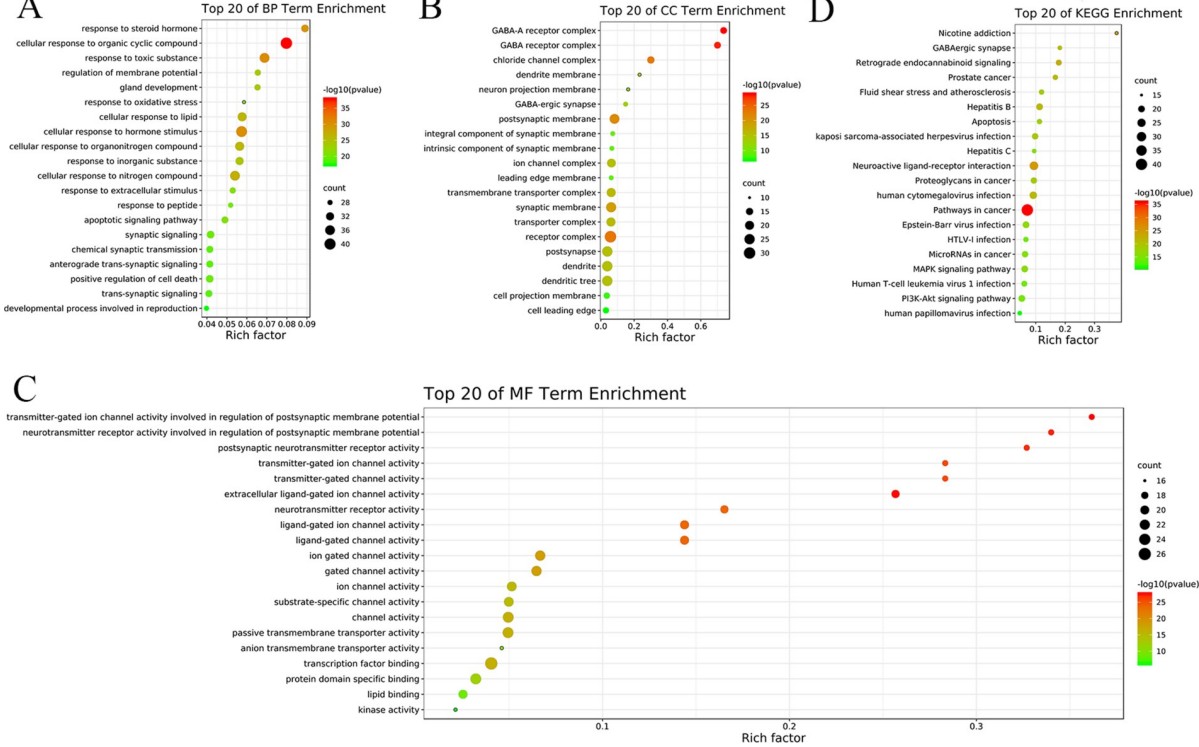

**Fig 6. GO and KEGG analysis of targets.** (A) GO biological process terms. (B) GO cellular component terms. (C) GO molecular function terms. (D) KEGG pathway. The size of each dot corresponds to the number of genes annotated in the entry, and the color of each dot corresponds to the corrected p-value.

to lipids (GO:0071396) (**Fig 6A**), cellular components (CCs) in receptor complexes (GO: 0043235), postsynaptic (GO: 0098794), dendrites (GO: 0030425), dendritic tree (GO: 0097447), synaptic membrane (GO: 0097060) (**Fig 6B**), molecular function (MF) in transcription factor binding (GO: 0008134), channel activity (GO: 0015267), passive transmembrane transporter activity (GO: 0022803), protein domain-specific binding (GO: 0019904), and ion gated channel activity (GO: 0022839) (**Fig 6C**). According to the count, the top 10 BPs, top 10 CCs, and top 10 MFs are presented in **Table 3**.

**Table 3. GO enrichment analysis.**

| Category | Term | Count | P value |
|---|---|---|---|
| | GO:0071407~cellular response to organic cyclic compound | 41 | 1.44229E-38 |
| | GO:0032870~cellular response to hormone stimulus | 39 | 4.01273E-31 |
| | GO:0009636~response to toxic substance | 36 | 1.8724E-31 |
| | GO:1901699~cellular response to nitrogen compound | 36 | 8.96277E-28 |
| BP | GO:0071396~cellular response to lipid | 34 | 4.05399E-27 |
| | GO:0071417~cellular response to organonitrogen compound | 34 | 8.29749E-27 |
| | GO:0010035~response to inorganic substance | 32 | 3.30251E-25 |
| | GO:0048545~response to steroid hormone | 31 | 1.71101E-30 |
| | GO:0099536~synaptic signaling | 31 | 1.28938E-20 |
| | GO:0010942~positive regulation of cell death | 31 | 1.75892E-20 |
| | GO:0043235~receptor complex | 30 | 5.57143E-24 |

(*Continued*)

**Table 3.** (Continued)

| Category | Term | Count | P value |
|---|---|---|---|
| | GO:0098794~post synapse | 25 | 5.3657E-16 |
| | GO:0030425~dendrite | 25 | 5.76475E-16 |
| | GO:0097447~dendritic tree | 25 | 6.19191E-16 |
| CC | GO:0097060~synaptic membrane | 24 | 5.88958E-20 |
| | GO:0045211~postsynaptic membrane | 23 | 6.55646E-22 |
| | GO:1902495~transmembrane transporter complex | 20 | 8.97726E-17 |
| | GO:1990351~transporter complex | 20 | 1.44266E-16 |
| | GO:0034702~ion channel complex | 19 | 3.68031E-07 |
| | GO:0034707~chloride channel complex | 15 | 1.38207E-23 |
| | GO:0008134~transcription factor binding | 26 | 5.25479E-17 |
| | GO:0015267~channel activity | 23 | 5.2142E-17 |
| | GO:0022803~passive transmembrane transporter activity | 23 | 5.46442E-17 |
| | GO:0019904~protein domain specific binding | 23 | 4.86877E-13 |
| MF | GO:0022839~ion gated channel activity | 22 | 5.04078E-19 |
| | GO:0022836~gated channel activity | 22 | 9.55578E-19 |
| | GO:0005216~ion channel activity | 21 | 6.33001E-16 |
| | GO:0022838~substrate-specific channel activity | 21 | 1.18187E-15 |
| | GO:0008289~lipid binding | 20 | 1.21686E-09 |
| | GO:0015276~ligand-gated ion channel activity | 20 | 4.02787E-24 |

## KEGG pathway enrichment analysis

To elucidate the crucial signaling pathways of YSTLF in the treatment of oligoasthenozoospermia, the 134 potential targets were input into Metascape for KEGG pathway enrichment analysis. The top 20 pathways were screened based on the parameters of counts as well as in combination with P-values (Table 4), including the MAPK signaling pathway (hsa04010),

**Table 4. KEGG pathway enrichment analysis based on the YSTLF-oligoasthenozoospermia network (top 20 with count).**

| Pathway | Genes | Count | P value |
|---|---|---|---|
| Pathways in cancer | AKR1B1, BIRC5, AR, CCND1, BCL2, CASP3, CASP7, CASP8, CASP9, NQO1, EGFR, ELK1, ERBB2, ESR1, ESR2, FASN, FOS, GSK3B, GSTM1, GSTP1, HIF1A, IGF2, IKBKB, IL6, IL13, MDM2, MYC, NFE2L2, NFKBIA, PPARG, PRKCA, MAPK8, PTGER4, PTGS2, RAF1, RARA, RB1, RELA, RXRA, VEGFA, RASSF1 | 41 | 1.3986E-36 |
| Neuroactive ligand-receptor interaction | ADRA1A, CALCR, CHRNA3, CHRNA7, CHRNB4, DRD2, GABRA1, GABRA2, GABRA3, GABRA4, GABRA5, GABRA6, GABRB1, GABRB2, GABRB3, GABRD, GABRE, GABRG1, GABRG3, GLRA1, NR3C1, PRLR, PRSS1, PTGER4, TRPV1, GABRQ | 26 | 3.3424E-26 |
| human cytomegalovirus infection | CCND1, CASP3, CASP8, CASP9, EGFR, ELK1, FASN, GSK3B, IKBKB, IL1B, IL6, MDM2, MYC, NFKBIA, PRKCA, PTGER4, PTGS2, RAF1, RB1, RELA, TNF, VEGFA | 22 | 4.9049E-22 |
| Hepatitis B | BIRC5, CCND1, BCL2, CASP3, CASP8, CASP9, ELK1, FASN, FOS, IKBKB, IL6, MYC, NFKBIA, PCNA, PRKCA, MAPK8, RAF1, RB1, RELA, TNF | 20 | 6.0125E-22 |
| Epstein-Barr virus infection | CCND1, BCL2, CASP3, CASP8, CASP9, CD44, FASN, GSK3B, HSPB1, ICAM1, IKBKB, IL6, MDM2, MYC, NFKBIA, MAPK8, PSMD3, RB1, RELA, TNF | 20 | 2.1281E-17 |
| MAPK signaling pathway | CACNA1C, CASP3, EGFR, ELK1, ERBB2, ERBB3, FASN, FOS, HSPB1, IGF2, IKBKB, IL1B, IL6, MYC, PRKCA, MAPK8, RAF1, RELA, TNF, VEGFA | 20 | 7.9572E-17 |
| PI3K-Akt signaling pathway | CCND1, BCL2, CASP9, EGFR, ERBB2, ERBB3, GSK3B, IGF2, IKBKB, IL6, MCL1, MDM2, MYC, NOS3, PRKCA, PRLR, RAF1, RELA, RXRA, VEGFA | 20 | 2.1556E-15 |
| kaposi sarcoma-associated herpesvirus infection | CCND1, CASP3, CASP8, CASP9, FASN, FOS, GSK3B, HIF1A, ICAM1, IKBKB, IL6, MYC, NFKBIA, MAPK8, PTGS2, RAF1, RB1, RELA, VEGFA | 19 | 1.0748E-19 |

(*Continued*)

**Table 4.** (Continued)

| Pathway | Genes | Count | P value |
|---|---|---|---|
| Proteoglycans in cancer | CCND1, CASP3, CAV1, CD44, EGFR, ELK1, ERBB2, ERBB3, ESR1, HIF1A, IGF2, MDM2, MYC, PLAU, PRKCA, RAF1, SLC9A1, TNF, VEGFA | 19 | 2.5601E-19 |
| MicroRNAs in cancer | CCND1, BCL2, CASP3, CD44, CYP1B1, EGFR, ERBB2, ERBB3, IKBKB, MCL1, MDM2, MYC, PLAU, PRKCA, PTGS2, RAF1, VEGFA, TP63, RASSF1 | 19 | 3.659E-16 |
| Retrograde endocannabinoid signaling | CACNA1C, GABRA1, GABRA2, GABRA3, GABRA4, GABRA5, GABRA6, GABRB1, GABRB2, GABRB3, GABRD, GABRE, GABRG1, GABRG3, PRKCA, MAPK8, PTGS2, GABRQ | 18 | 1.4255E-23 |
| Human T-cell leukemia virus 1 infection | CCND1, CHEK1, ELK1, FOS, GSK3B, ICAM1, IKBKB, IL6, MYC, NFKBIA, PCNA, PRKCA, MAPK8, RB1, RELA, TNF, VCAM1, CHEK2 | 18 | 4.6747E-15 |
| Prostate cancer | AKR1B1, AR, CCND1, BCL2, CASP9, EGFR, ERBB2, GSK3B, GSTP1, IKBKB, MDM2, NFKBIA, PLAU, RAF1, RB1, RELA, SRD5A2 | 17 | 8.6806E-22 |
| Fluid shear stress and atherosclerosis | BCL2, CAV1, NQO1, FOS, GSTM1, GSTP1, ICAM1, IKBKB, IL1B, NFE2L2, NOS3, MAPK8, RELA, TNF, VCAM1, VEGFA, NOX1 | 17 | 3.1072E-19 |
| Apoptosis | PARP1, BIRC5, BCL2, CASP3, CASP7, CASP8, CASP9, CTSD, FASN, FOS, IKBKB, MCL1, NFKBIA, MAPK8, RAF1, RELA, TNF | 17 | 9.0654E-19 |
| HTLV-I infection | CCND1, CHEK1, ELK1, FOS, GSK3B, ICAM1, IKBKB, IL6, MYC, NFKBIA, PCNA, MAPK8, RB1, RELA, TNF, VCAM1, CHEK2 | 17 | 6.9835E-15 |
| GABAergic synapse | CACNA1C, GABRA1, GABRA2, GABRA3, GABRA4, GABRA5, GABRA6, GABRB1, GABRB2, GABRB3, GABRD, GABRE, GABRG1, GABRG3, PRKCA, GABRQ | 16 | 3.265E-21 |
| Hepatitis C | CCND1, CASP3, CASP8, CASP9, EGFR, FASN, GSK3B, IKBKB, MYC, NFKBIA, MAPK8, RAF1, RB1, RELA, RXRA, TNF | 16 | 1.6607E-16 |
| human papillomavirus infection | CCND1, CASP3, CASP8, EGFR, FASN, GSK3B, IKBKB, MDM2, PRKCA, PTGER4, PTGS2, RAF1, RB1, RELA, TNF, VEGFA | 16 | 1.4763E-11 |
| Nicotine addiction | CHRNA7, GABRA1, GABRA2, GABRA3, GABRA4, GABRA5, GABRA6, GABRB1, GABRB2, GABRB3, GABRD, GABRE, GABRG1, GABRG3, GABRQ | 15 | 2.5707E-25 |

PI3K/Akt signaling pathway (hsa04151), and apoptosis (hsa04210). Excluding the above, the corresponding common signaling pathways also focused on the TNF signaling pathway (hsa04668), AGE-RAGE signaling pathway in diabetic complications (hsa04933), and IL-17 signaling pathway (hsa04657), which had lower P-values. The KEGG pathways were visualized via the Omcishare website (https://www.omicshare.com/) based on their corresponding counts (**Fig 6D**).

A disease-formula-compound-target-pathway network was constructed using the Cytoscape 3.7.2 platform (**Fig 7**). Based on the above results, the PI3K/Akt, MAPK and apoptosis signaling pathways were found to be the most important signaling pathways during the process of treating oligoasthenozoospermia with YSTLF. The targets related to these three signaling pathways are shown in Figs 8–10, which were retrieved from the KEGG database (http://www.kegg.jp/kegg/mapper.html).

## Discussion

Oligoasthenozoospermia is a multifactorial disease that is associated with a variety of proteins or pathways during its occurrence and progression, such as calcium- and integrin-binding protein-1 (CIB1) and the TGF-β1/Smad signaling pathway [28, 29]. On account of their multi-component compositions, Chinese herbal medicines may exhibit a wide range of pharmacological activities with the characteristics of multiple targets and multiple pathways, which may be beneficial to the treatment of oligoasthenozoospermia [30]. YSTLF has achieved satisfactory therapeutic effects in the treatment of oligoasthenozoospermia. According to the literature, YSTLF with or without minimally invasive surgery in the treatment of varicocele infertility could improve the sperm motility rate, sperm linear motion velocity, sperm concentration,

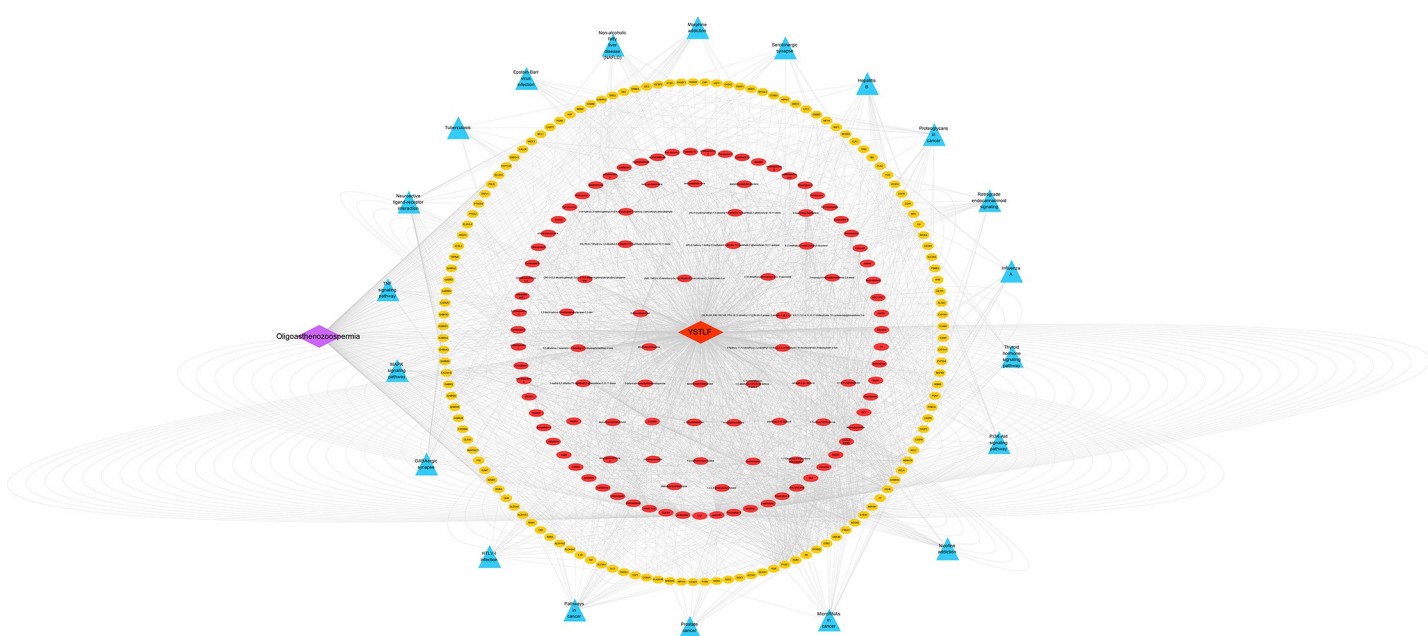

**Fig 7. Disease-formula-compound-target-pathway network.** Purple rhombuses represent disease, red rhombuses represent formulae, red ellipses represent the chemical compounds in YSTLF related to the common targets, yellow polygons represent the common targets from the chemical compounds and oligoasthenozoospermia, and blue triangles represent the main biological pathways.

sperm DNA integrity and spousal pregnancy rate [6, 31]. However, the underlying mechanisms of YSTLF in the treatment of oligoasthenozoospermia remain unclear and need to be elucidated through an in-depth systematic study at the molecular level. Based on the theories of molecular biology, systems biology and pharmacology, network pharmacology can construct intricate interaction networks according to the bioactive compounds, target molecules, and biological functions and may clarify the potential molecular mechanisms of complicated Chinese medicine (CM) formulae in diseases. The network pharmacology approach may be a promising systematic mechanistic research strategy for CM formula studies. Thus, in the current study, the above approach was adopted to illuminate the underlying pharmacological mechanisms of YSTLF in oligoasthenozoospermia treatment. The components in YSTLF with OB≥30% and DL≥0.18 were deemed pharmacokinetically active and are probably absorbed and distributed in the body. Accordingly, the 106 bioactive ingredients (the top 10 compounds with high degrees were quercetin, luteolin, kaempferol, crocetin, D-mannitol, ursolic acid, tanshinone iia, isorhamnetin, anhydroicaritin and beta-sitosterol) and the 134 potential targets (the 10 hub targets were TNF, RELA, CCND1, ESR1, RXRA, NFKBIA, GSK3B, NR3C1, MAPK8, IL6) of YSTLF for oligoasthenozoospermia treatment were predicted, which could likely reveal the potential pharmacological mechanisms of YSTLF.

According to the literature, factors affecting the count and quality of sperm include oxidative stress, inflammatory reactions, immunologic derangement and DNA damage to sperm, etc. In this study, quercetin was the most significant compound, which is closely related to semen quality. Quercetin, a compound belonging to the flavonol class, has extensive pharmacological actions, such as antioxidant and anti-inflammatory effects [32]. In vitro experiments have shown that adding quercetin to semen can elevate the total antioxidant potential and ameliorate lipid peroxidation during cryopreservation to improve sperm motility, acrosome integrity, plasma membrane integrity, mitochondrial activity and chromatin condensation [33]. Diao et al. demonstrated that quercetin can significantly boost sperm motility in

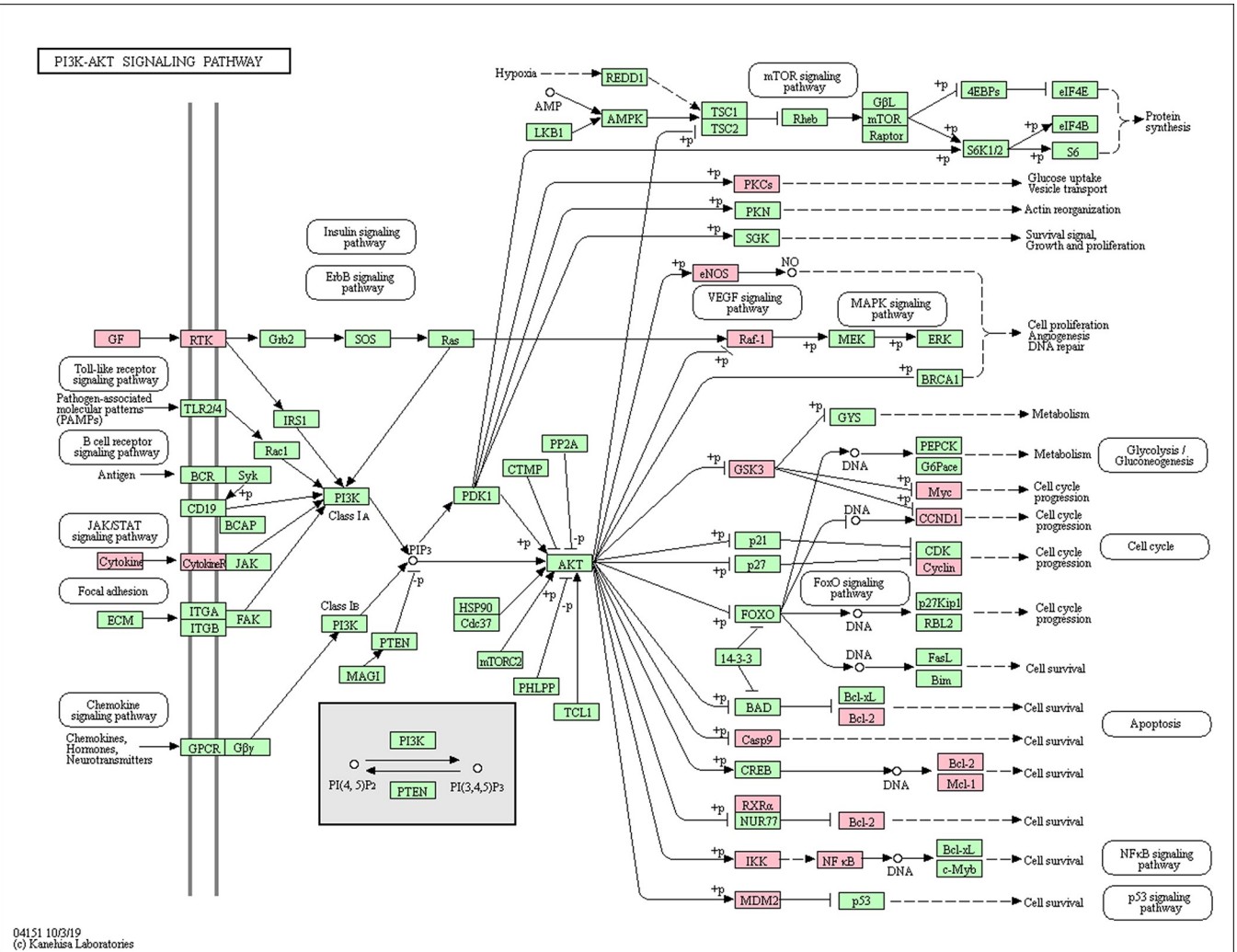

**Fig 8. PI3K-AKT signaling pathway.** The red rectangle represents the targets related to the core component-target-pathway network.

leukocytospermic patients, and the mechanism could be that quercetin can reduce the levels of HO and the content of mtDNA and increase the contents of CytB and NADH5 in sperm to display intensive antioxidant activity against ROS-mediated sperm damage [34]. In addition, luteolin and kaempferol, which are also flavonoids with antioxidant effects, can improve sperm quality by increasing the activity of superoxide dismutase (SOD) and glutathione peroxidase (GPx) or repairing sperm DNA damage [35, 36]. Crocetin, a carotenoid, also has strong antioxidant activity, especially in the suppression of lipid peroxidation reactions; on the other hand, it can decrease testicular apoptosis by reducing caspase 3 activity, thus promoting spermatogenic function [37]. Isorhamnetin-containing flavonoids and the phytosterol derivative beta-sitosterol can both alleviate oxidative stress and apoptosis in sperm [38, 39]. A network pharmacology approach together with all of these studies can predict that YSTLF may improve the count and quality of sperm by exerting antioxidant and antiapoptotic effects, thus improving male fertility.

Among the top 10 hub targets, many genes were found to be involved in spermatogenesis, sperm motility and sperm morphology, which play an important role in the therapeutic action

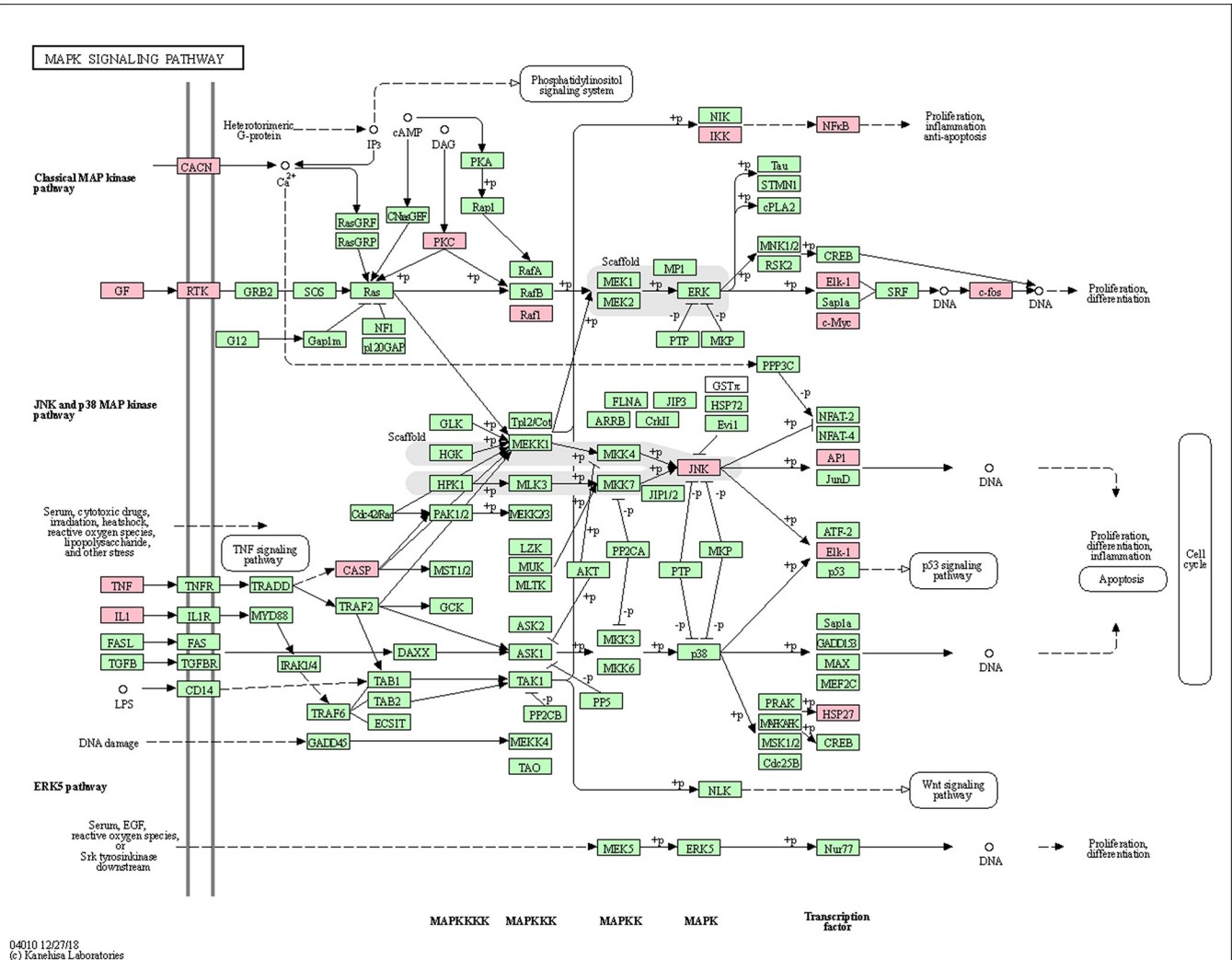

**Fig 9. MAPK signaling pathway.** The red rectangle represents the targets related to the core component-target-pathway network.

of YSTLF for oligoasthenozoospermia treatment. As inflammatory cytokines, increased TNF and IL6 in the male reproductive tract could jeopardize spermatogenesis as well as the functions of sperm and ejaculation, which are closely related to fertility problems [40, 41]. In addition, TNF and IL6 could activate apoptotic mechanisms in human spermatozoa [42–44]. As an inflammatory mediator, the content of NFKBIA may decrease, while cytokines such as IL6 may increase in the epididymitis. The above early inflammatory signaling events could adversely affect male fertility [45]. Cyclin D1 (CCND1), a cell cycle factor, is associated with cell cycle progression and spermatogenesis [46]. ESR1 is expressed throughout the reproductive tracts of adult male mice, and mutation to ESR1 can reduce sperm viability without affecting testicular size or sperm count, thus impairing male fertility [47]. Furthermore, polymorphisms in exon 4 (LBD) of ESR1 also have a close association with male infertility [48]. Another study reported that ESR1 is essential for sperm survival and maturation during epididymal storage because it is involved in the reabsorption of fluid in the cavity during sperm transport from the testis to the epididymis head. In addition, ESR1 knockout causes reduced sperm content in the epididymis and decreased sperm motility [49]. The NR3C1

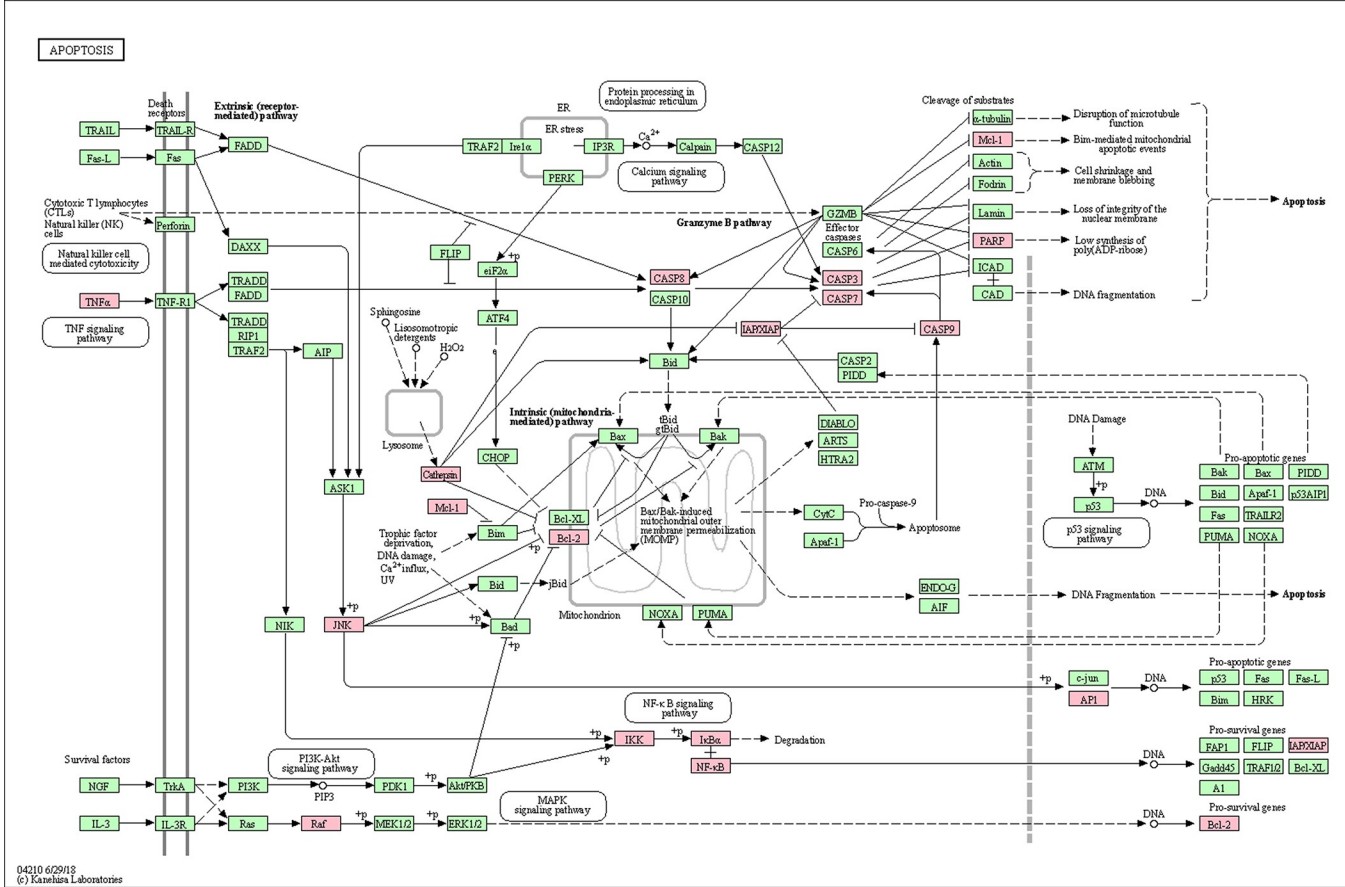

**Fig 10. Apoptosis signaling pathway.** The red rectangle represents the targets related to the core component-target-pathway network.

protein, which is mainly distributed in peritubular cells, Sertoli cells, Leydig cells and spermatogonia of the adult testis, has been shown to be strongly associated with sperm motility [50]. The C-Jun N-terminal kinase MAPK8, which is a key gene in the Aktsignaling pathway, is considered to be associated with sperm apoptosis [51]. Based on the above results, YSTLF may increase sperm count and improve sperm motility by regulating cell proliferation, survival, and apoptosis and the inflammatory response of reproduction-related cells in males.

In the present study, KEGG pathway enrichment analysis showed that there were multiple signaling pathways involved in the treatment of oligoasthenozoospermia with YSTLF. Among them, the PI3K-Akt, MAPK and apoptosis pathways, in which YSTLF-associated hub targets were enriched, are closely related to oligoasthenozoospermia. The PI3K/Akt signaling pathway can participate in many important processes of male reproduction, such as the proliferation, differentiation, growth and apoptosis of spermatogonia and regulation of the hypothalamic-pituitary-gonad (HPG) axis during spermatogenesis [52]. Hypoxia and oxidative stress can inhibit the PI3K/Akt signaling pathway, reducing the level of Akt phosphorylation and inducing cell apoptosis [53]. Other studies have demonstrated that activation of the PI3K/Akt pathway may protect testicular cells from apoptosis in the offspring of mice after direct maternal exposure to di-(2-ethylhexyl) phthalate (DEHP) [52, 54]. In another study, suppression of the PI3K/Akt pathway associated with oxidative stress may lead to aflatoxin $B_1$ ($AFB_1$)-induced testicular damage and promote autophagy [55]. MAPK, one of the members of the serine/threonine protein kinase family, is involved in many physiological processes, such as cell

proliferation, differentiation and growth, and P38, JNK and ERK, three protein genes in the MAPK pathway, are closely related to oligoasthenozoospermia [56]. The JNK and ERK1/2 MAPK signaling pathways, which are activated by di-N-butyl-phthalate (DBP), result in testicular injury, thus reducing sperm count by increasing apoptosis [57]. In addition, the P38 MAPK pathway could be activated by an abnormal arachidonic acid (AA) metabolic network, leading to decreased sperm motility [58]. Apoptosis plays a crucial role in spermatogenesis, so this pathway was predicted to be involved in the current study by the network pharmacology approach. NFKBIA, MAPK8, RELA, and TNF in the hub targets were enriched in the apoptosis signaling pathway. In summary, YSTLF may have therapeutic action in oligoasthenozoospermia mainly through antioxidative stress, antiapoptosis and anti-inflammation effects, which is basically consistent with the relevant research results of oligoasthenozoospermia in recent years. For instance, Morinda offcinalis–Lycium barbarum coupled-herbs (MOLBCH) can regulate the PI3K/Akt signaling pathway, prostate cancer, and the AGE-RAGE signaling pathway, regarded as the most representative pathways, by the core potential targets, the androgen receptor (AR), estrogen receptor (ESR1), mitogen-activated protein kinase 3 (MAPK3), RAC-alpha serine/threonine-protein kinase (AKT1), and glyceraldehyde-3-phosphate dehydrogenase (GAPDH) to alleviate apoptosis, promote male reproductive function, and reduce oxidant stress in the treatment of oligoasthenozoospermia [59].

However, the current study had some limitations. For instance, whether the functions of the pathways in this research were downregulated or upregulated was not clear. The potential targets of YSTLF in oligoasthenozoospermia treatment identified by the network pharmacology approach are only theoretical predictions and need to be verified by clinical and cell or animal experiments.

## Conclusions

In this study, the molecular mechanisms of YSTLF in oligoasthenozoospermia treatment were investigated by establishing multiple network models from a holistic viewpoint. It was demonstrated that the most effective compounds of YSTLF in the treatment of oligoasthenozoospermia may be quercetin, luteolin, kaempferol, crocetin, isorhamnetin and beta-sitosterol. The potential pharmacological actions of the above bioactive ingredients were closely related to antioxidative stress, antiapoptosis and anti-inflammation, with TNF, CCND1, ESR1, NFKBIA, NR3C1, MAPK8, and IL6 being possible targets. Furthermore, YSTLF may increase sperm count and motility mainly through the regulation of the PI3K/Akt, MAPK and apoptosis signaling pathways. The above study also suggested that network pharmacology predictions may be helpful tools to illustrate the molecular mechanisms of the Chinese herbal compound YSTLF in oligoasthenozoospermia treatment.

## Acknowledgments

All the authors of the manuscript are immensely grateful to the foundations for their valuable support.

## Author Contributions

**Data curation:** Yangdi Chen.

**Formal analysis:** Yangdi Chen.

**Funding acquisition:** Zixue Sun.

**Methodology:** Fanggang Bi.

**Project administration:** Yangdi Chen.

**Supervision:** Zixue Sun.

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
