## [Decision Letter · Decision Letter 0]

29 Apr 2021

PONE-D-21-02516

A Network Pharmacology Approach to the Underlying Mechanisms of Action of Yishen Tongluo Formula for the Treatment of Oligoasthenozoospermia

PLOS ONE

Dear Dr. Sun,

Thank you for submitting your manuscript to PLOS ONE. After careful consideration, we feel that it has merit but does not fully meet PLOS ONE’s publication criteria as it currently stands. Therefore, we invite you to submit a revised version of the manuscript that addresses the points raised during the review process.

After careful reviewing of this manuscripts by keen reviewers from the field, I think this manuscript can be published after addressing comments by reviewers. In specific, the manuscript need improvement in language and writing, preferably by a native English speaker. In addition, representing data need corrections and improvements as mentioned in comments.

We look forward to receiving your revised manuscript.

Kind regards,

Moustafa Elsayed El-Araby, Ph.D.

Academic Editor

PLOS ONE

Journal Requirements:

4. Please include your tables as part of your main manuscript and remove the individual files. Please note that supplementary tables should  be uploaded as separate "supporting information" files.

Additional Editor Comments:

Dear Dr. Zixue Sun (Author),

After careful reviewing of this manuscripts by keen reviewers from the field, I think this manuscript can be published after addressing comments by reviewers. In specific, the manuscript need improvement in language and writing, preferably by a native English speaker. In addition, representing data need corrections and improvements as mentioned in comments.

Reviewers' comments:

Reviewer's Responses to Questions

**Comments to the Author**

1. Is the manuscript technically sound, and do the data support the conclusions?

Reviewer #1: No

Reviewer #2: Yes

Reviewer #3: Yes

Reviewer #4: Yes

2. Has the statistical analysis been performed appropriately and rigorously? 

Reviewer #1: No

Reviewer #2: Yes

Reviewer #3: N/A

Reviewer #4: N/A

3. Have the authors made all data underlying the findings in their manuscript fully available?

Reviewer #1: Yes

Reviewer #2: Yes

Reviewer #3: Yes

Reviewer #4: Yes

4. Is the manuscript presented in an intelligible fashion and written in standard English?

Reviewer #1: No

Reviewer #2: Yes

Reviewer #3: Yes

Reviewer #4: No

5. Review Comments to the Author

Reviewer #1: This study is solely based on ‘putative’ targets, ‘potential’ targets/mechanisms retrieved from databases and analyzed to make connections. Spermatogenesis is one of the fundamental process that sustains life and, undoubtedly, there will be hundreds of molecular mechanisms regulating such processes. Mere bioinformatics analysis of ‘putative’ targets and linking them to ‘potential’ mechanisms does not warrant publication of this paper in PlosOne. Needless to comment on grammatical and typographical errors throughout the text.

Reviewer #2: The manuscript entitled “a network pharmacology approach to the underlying mechanisms of action of Yishen Tongluo formula for the treatment of oligoasthenozoospermia.” investigated the application of network pharmacology approach in exploring the potential mechanisms of YSTLF on the complex disease “oligoasthenozoospermia”. In my opinion, the current data provided robust evidences for the suggested mechanisms with respect to anti-oxidative stress, anti-apoptosis and anti-inflammation. The overall manuscript is well written in good English language with updated references. Nonetheless, authors should address the following minor considerations:

Minor Comments:

• Figures 4 & 6 - in the PDF version I have received from the journal - are NOT clear. I could hardly read the labels. This bad resolution is NOT suitable for publications. In my opinion, the original figures have to be enhanced to be not less than 900 dpi, so that they could be readable in PDF version.

• It is obvious for any reader that the words “we” & “our” were repeated in many sentences. I think it could be more convenient to rephrase such sentences to be in passive voice, which is more suitable for scientific writing.

Reviewer #3: Thank you for the opportunity to review the manuscript with the objective to utilize network pharmacology approach to elucidate chemical and pharmacological profiles of Yishen Tongluo formula (YSTLF) composed of seven herbs of traditional Chinese medicine origin and to explore the potential mechanisms of action of YSTLF on oligoasthenozoospermia.

The study is based on the theories of multi-directional pharmacology and systems biology, and how network pharmacology could be used to construct complex network models to study the biological or pharmacological properties of the multi-component formula and explore its physiological or pharmacological mechanisms. It was supported by elaborate data analysis, virtual computing modelling technology and network public databases. It identified the probable most effective compounds and bioactive ingredients of YSTLF in treating oligoasthenozoospermia, their possible mode of actions and suggested that network pharmacology prediction may be a helpful tool to illustrate the interactive relationship between multi-components, multi-targets and multi-pathways of bioactive compounds in traditional Chinese herbal medicine towards evaluating their compatibility and rationality.

The methods used were sufficiently described for another researcher to reproduce the study with the same or similar methods.

The study limitations and areas of further research were highlighted.

Reviewer #4: The methodology of the work to answer the research question was correctly presented and carried out, appropriate to make the conclusions to be determined in your work.

I would make the following recommendations:

1. Language editing.

2. Be more specific with the results obtained in the clinical trial using YSTLF in conjunction with the surgical procedure in the treatment of oligoasthenozoospermia.

3. Generate a discussion that could be enriched with the results presented by the recent publication with the title: Network pharmacology integrated molecular docking reveals the bioactive components and potential targets of Morinda officinalis – Lycium barbarum coupled-herbs against oligoasthenozoospermia.

4. Possibly the development of a diagram that represents the possible mechanisms involved.

6. PLOS authors have the option to publish the peer review history of their article (what does this mean?). If published, this will include your full peer review and any attached files.

Reviewer #1: **Yes: **Sathya Velmurugan

Reviewer #2: No

Reviewer #3: **Yes: **Taiwo Oyelade

Reviewer #4: **Yes: **Mario Alberto Garza-Garza

---

## [Author Response · Author response to Decision Letter 0]

20 May 2021

Dear Editor：

Thank you for your letter and for the reviewers’ comments concerning our manuscript entitled “A Network Pharmacology Approach to the Underlying Mechanisms of Action of Yishen Tongluo Formula for the Treatment of Oligoasthenozoospermia” (ID: PONE-D-21-02516). Those comments are all valuable and very helpful for revising and improving our paper, as well as the important guiding significance to our manuscript. We have studied comments carefully and have made corrections which we hope meet with approval. In this revised version, changes to our manuscript within the document were all highlighted by using red colored text. Point-to-point responses to the reviewers are listed below this letter.

Additional Editor Comments:

Dear Dr. Zixue Sun (Author),

After careful reviewing of this manuscripts by keen reviewers from the field, I think this manuscript can be published after addressing comments by reviewers. In specific, the manuscript need improvement in language and writing, preferably by a native English speaker. In addition, representing data need corrections and improvements as mentioned in comments.

Reply: This document certifies that the manuscript “A Network Pharmacology Approach to the Underlying Mechanisms of Action of Yishen Tongluo Formula for the Treatment of Oligoasthenozoospermia” prepared by the authors Yangdi Chen, Fanggang Bi, Zixue Sun was edited for proper English language, grammar, punctuation, spelling, and overall style by one or more of the highly qualified native English-speaking editors at SNAS. This certificate was issued on May 13, 2021 and may be verified on the SNAS website using the verification code F84C-D686-DCDD-B7C0-AF45.

Responds to the reviewer’s comments:

Reviewer #1: This study is solely based on ‘putative’ targets, ‘potential’ targets/mechanisms retrieved from databases and analyzed to make connections. Spermatogenesis is one of the fundamental process that sustains life and, undoubtedly, there will be hundreds of molecular mechanisms regulating such processes. Mere bioinformatics analysis of ‘putative’ targets and linking them to ‘potential’ mechanisms does not warrant publication of this paper in PlosOne. Needless to comment on grammatical and typographical errors throughout the text.

Reply: Thank you for your comments. This study, which is based on ‘putative’ targets, ‘potential’ targets/mechanisms retrieved from databases and analyzed to make connections, has a limitation, and we also point it out at the end of the article. However, network pharmacological methods can identify the probable most effective compounds and bioactive ingredients of YSTLF in treating oligoasthenozoospermia, their possible mode of actions and suggested that network pharmacology prediction may be a helpful tool to illustrate the interactive relationship between multi-components, multi-targets and multi-pathways of bioactive compounds in traditional Chinese herbal medicine towards evaluating their compatibility and rationality. Xue Bai et al.’s research(Bai X, Tang Y, Li Q, Chen Y, Liu D, Liu G, et al. Network pharmacology integrated molecular docking reveals the bioactive components and potential targets of Morinda officinalis-Lycium barbarum coupled-herbs against oligoasthenozoospermia. Sci Rep. 2021;11(1):2220. Epub 2021/01/28. doi: 10.1038/s41598-020-80780-6. PubMed PMID: 33500463; PubMed Central PMCID: PMCPMC7838196.), which was published in the journal Scientific Reports, demonstrated that Morinda offcinalis–Lycium barbarum coupled-herbs (MOLBCH) alleviated apoptosis, promoted male reproductive function, and reduced oxidant stress in the treatment of OA. Ohioensin-A, quercetin, beta-sitosterol and sitosterol were the key bioactive components. Androgen receptor (AR), Estrogen receptor (ESR1), Mitogen-activated protein kinase 3 (MAPK3), RAC-alpha serine/threonine-protein kinase (AKT1), Glyceraldehyde-3-phosphate dehydrogenase (GAPDH) were the core potential targets. PI3K/Akt signaling pathway, prostate cancer, AGE-RAGE signaling pathway in diabetic complications were the most representative pathways. Their research methods are almost the same as ours. So we think that our research has merit. These observations provide a reliable basis for future experiments and deeper insight into the pathogenesis of oligoasthenozoospermia, and can develop new therapeutic instructions to treat oligoasthenozoospermia. 

Reviewer #2: 

Comment 1. Figures 4 & 6 - in the PDF version I have received from the journal - are NOT clear. I could hardly read the labels. This bad resolution is NOT suitable for publications. In my opinion, the original figures have to be enhanced to be not less than 900 dpi, so that they could be readable in PDF version. 

Reply and modifications: Thank you for your valuable comments. However, PLOS ONE's figure file requirements are 300 – 600 dpi in resolution, <10 MB in file size. We have endeavoured to revise Figures 4 & 6 to meet the publication requirements and uploaded new Figures 4 & 6.

Comment 2. It is obvious for any reader that the words “we” & “our” were repeated in many sentences. I think it could be more convenient to rephrase such sentences to be in passive voice, which is more suitable for scientific writing.

Reply and modifications: In the revised version, we have changed the active voice to the passive voice to be more suitable for scientific writing.

Reviewer #3: Thank you for the opportunity to review the manuscript with the objective to utilize network pharmacology approach to elucidate chemical and pharmacological profiles of Yishen Tongluo formula (YSTLF) composed of seven herbs of traditional Chinese medicine origin and to explore the potential mechanisms of action of YSTLF on oligoasthenozoospermia.

The study is based on the theories of multi-directional pharmacology and systems biology, and how network pharmacology could be used to construct complex network models to study the biological or pharmacological properties of the multi-component formula and explore its physiological or pharmacological mechanisms. It was supported by elaborate data analysis, virtual computing modelling technology and network public databases. It identified the probable most effective compounds and bioactive ingredients of YSTLF in treating oligoasthenozoospermia, their possible mode of actions and suggested that network pharmacology prediction may be a helpful tool to illustrate the interactive relationship between multi-components, multi-targets and multi-pathways of bioactive compounds in traditional Chinese herbal medicine towards evaluating their compatibility and rationality.

The methods used were sufficiently described for another researcher to reproduce the study with the same or similar methods.

The study limitations and areas of further research were highlighted.

Reply: Thank you very much for your comments. Thank you for your affirmation and support for our research.

Reviewer #4: 

Comment 1. Language editing.

Reply: This document certifies that the manuscript “A Network Pharmacology Approach to the Underlying Mechanisms of Action of Yishen Tongluo Formula for the Treatment of Oligoasthenozoospermia” prepared by the authors Yangdi Chen, Fanggang Bi, Zixue Sun was edited for proper English language, grammar, punctuation, spelling, and overall style by one or more of the highly qualified native English-speaking editors at SNAS. This certificate was issued on May 13, 2021 and may be verified on the SNAS website using the verification code F84C-D686-DCDD-B7C0-AF45.

Comment 2. Be more specific with the results obtained in the clinical trial using YSTLF in conjunction with the surgical procedure in the treatment of oligoasthenozoospermia.

Reply and modifications: We have added the data of the above research result in the paper (Page 5, line 99) and cited literatures in the discussion section (Page 15, lines 316-319) in the file labeled 'Revised Manuscript with Track Changes'.

Comment 3. Generate a discussion that could be enriched with the results presented by the recent publication with the title: Network pharmacology integrated molecular docking reveals the bioactive components and potential targets of Morinda officinalis – Lycium barbarum coupled-herbs against oligoasthenozoospermia.

Reply and modifications: Thank you for your suggestion. We have cited this literature in the discussion section (Page 20, lines 425-432) to make our study more persuasive.

Comment 4. Possibly the development of a diagram that represents the possible mechanisms involved.

Reply and modifications: Thank you for your suggestion. We have added diagrams, which show the targets related to these three most important signaling pathways (PI3K/Akt, MAPK and Apoptosis signaling pathways) during the process of YSTLF in treating oligoasthenozoospermia. The above can represent the possible mechanisms involved.

If there are any other modifications we could make, we would like very much to modify them and we really appreciate your help. Thank you very much for your help.

Best wishes!

Zixue Sun 

Department of Reproductive Medicine, Henan Province Hospital of Traditional Chinese Medicine (The Second Affiliated Hospital of Henan University of Chinese Medicine)

No.6 Dongfeng Road, Jinshui District, Zhengzhou, China, 450002

---

## [Decision Letter · Decision Letter 1]

25 May 2021

A Network Pharmacology Approach to  Determine the Underlying Mechanisms of Action of Yishen Tongluo Formula for the Treatment of Oligoasthenozoospermia

PONE-D-21-02516R1

Dear Dr. Sun,

We’re pleased to inform you that your manuscript has been judged scientifically suitable for publication and will be formally accepted for publication once it meets all outstanding technical requirements.

Kind regards,

Moustafa E. El-Araby, Ph.D.

Academic Editor

PLOS ONE

Additional Editor Comments (optional):

Thank you for making corrections and responding to comments. The manuscript can be published in PLOS ONE.

Reviewers' comments:

Reviewer's Responses to Questions

**Comments to the Author**

1. If the authors have adequately addressed your comments raised in a previous round of review and you feel that this manuscript is now acceptable for publication, you may indicate that here to bypass the “Comments to the Author” section, enter your conflict of interest statement in the “Confidential to Editor” section, and submit your "Accept" recommendation.

Reviewer #2: All comments have been addressed

2. Is the manuscript technically sound, and do the data support the conclusions?

Reviewer #2: Yes

3. Has the statistical analysis been performed appropriately and rigorously? 

Reviewer #2: Yes

4. Have the authors made all data underlying the findings in their manuscript fully available?

Reviewer #2: Yes

5. Is the manuscript presented in an intelligible fashion and written in standard English?

Reviewer #2: Yes

6. Review Comments to the Author

Reviewer #2: The authors have responded to comments point to point.

The manuscript is now suitable for publication in PONE.

7. PLOS authors have the option to publish the peer review history of their article (what does this mean?). If published, this will include your full peer review and any attached files.

Reviewer #2: No

---

## [Editor Report · Acceptance letter]

10 Jun 2021

PONE-D-21-02516R1 

A Network Pharmacology Approach to Determine the Underlying Mechanisms of Action of Yishen Tongluo Formula for the Treatment of Oligoasthenozoospermia 

Dear Dr. Sun:

I'm pleased to inform you that your manuscript has been deemed suitable for publication in PLOS ONE. Congratulations! Your manuscript is now with our production department. 

Kind regards, 

on behalf of

Dr. Moustafa Elsayed El-Araby 

Academic Editor

PLOS ONE